# Specific Activation of Yamanaka Factors via HSF1 Signaling in the Early Stage of Zebrafish Optic Nerve Regeneration

**DOI:** 10.3390/ijms24043253

**Published:** 2023-02-07

**Authors:** Kayo Sugitani, Takumi Mokuya, Shuichi Homma, Minami Maeda, Ayano Konno, Kazuhiro Ogai

**Affiliations:** 1Department of Clinical Laboratory Science, Graduate School of Medical Science, Kanazawa University, 5-11-80 Kodatsuno, Kanazawa 920-0942, Japan; 2AI Hospital/Macro Signal Dynamics Research and Development Center, 5-11-80 Kodatsuno, Kanazawa 920-0942, Japan

**Keywords:** HSF1, Klf4, Oct4, Sox2, Yamanaka factors, retina, optic nerve regeneration, zebrafish

## Abstract

In contrast to the case in mammals, the fish optic nerve can spontaneously regenerate and visual function can be fully restored 3–4 months after optic nerve injury (ONI). However, the regenerative mechanism behind this has remained unknown. This long process is reminiscent of the normal development of the visual system from immature neural cells to mature neurons. Here, we focused on the expression of three Yamanaka factors (Oct4, Sox2, and Klf4: OSK), which are well-known inducers of induced pluripotent stem (iPS) cells in the zebrafish retina after ONI. mRNA expression of OSK was rapidly induced in the retinal ganglion cells (RGCs) 1–3 h after ONI. Heat shock factor 1 (HSF1) mRNA was most rapidly induced in the RGCs at 0.5 h. The activation of OSK mRNA was completely suppressed by the intraocular injection of HSF1 morpholino prior to ONI. Furthermore, the chromatin immunoprecipitation assay showed the enrichment of OSK genomic DNA bound to HSF1. The present study clearly showed that the rapid activation of Yamanaka factors in the zebrafish retina was regulated by HSF1, and this sequential activation of HSF1 and OSK might provide a key to unlocking the regenerative mechanism of injured RGCs in fish.

## 1. Introduction

Neurons in the mammalian central nervous system (CNS) cannot regenerate after nerve injury and eventually die, whereas neurons in the fish CNS can regenerate and fully recover CNS function [1,2,3]. Since the work of Sperry in the 1950s, the fish visual system has been the most popular model of CNS regeneration [4,5,6,7,8,9]. For the past 20 years, we have examined the regeneration of fish optic nerves from nerve crush to the recovery of visual function by using modern neurobiological tools such as immunohistochemistry [10,11,12], cell and tissue culture systems [13,14,15], and three-dimensional image processing systems for behavioral analysis [11,16,17]. The obtained results revealed that fish optic nerve regeneration includes (i) an early preparation period at 0–4 days; (ii) a middle neurite outgrowth period at 5–30 days; and (iii) a late synaptic refinement period at 1–4 months after optic nerve injury (ONI). We were particularly interested in the early period (0–4 days) because molecular events arising in this period are the most mysterious and important for resolving the regenerative mechanism of adult fish retinal ganglion cells (RGCs) after ONI.

In the past 10 years, we have applied molecular genetics in the search for genes upregulated at this early stage using zebrafish. Some cell survival-related and anti-apoptotic factors were found to be induced in RGCs within 1–4 days after ONI such as insulin-like growth factor-I (IGF-I), Bcl-2, phospho-Akt (p-Akt), and phospho-Bad (p-Bad) [17,18]. Similarly, molecules such as purpurin [19], neuroglobin [20,21], and cellular factor XIII A subunit (cFXIII-A) [12,15] were induced in RGCs to activate neurite outgrowth. Heat shock factor 1 (HSF1) was most rapidly induced in RGCs at 0.5 h after ONI [22,23,24]. These molecules were shown to function to maintain the viability of the injured RGCs and activate neural budding in preparation for promoted neurite elongation in the next stage [22,23,24].

However, why adult fish RGCs can regenerate after ONI has remained unclear. In consideration of the similarity between the regenerative process of the fish optic nerve and the normal development of the visual system in embryogenesis, we hypothesize that the injured fish RGCs are initialized to immature RGCs as soon as possible at the early stage after ONI. These immature RGCs can easily regenerate, regrow their axons, and restore visual function.

In the present study, we investigated the expression of Yamanaka factor genes after ONI in zebrafish. The term “Yamanaka factors” originally referred to four transcription factors, Oct4, Sox2, Klf4, and c-Myc, which have the effect of inducing somatic cells to become induced pluripotent stem (iPS) cells [25,26]. Of these four Yamanaka factors, c-Myc was reported to not necessarily be essential for initiating cell reprogramming [27,28,29]. Therefore, we evaluated the expression of three Yamanaka factors, Oct4, Sox2, and Klf4 (OSK), and their relationship to optic nerve regeneration. We also focused on the interaction between the expression of OSK and HSF1 as the fastest acute-phase response molecule after ONI.

## 2. Results

### 2.1. Rapid Increase of HSF1 Gene Expression in Zebrafish Retina after ONI

We performed real-time PCR using gene-specific primers to examine how the expression of the *HSF1* gene changes in the retina after optic nerve crush. The upregulation of *HSF1* mRNA started at 0.5 h after ONI, peaked at 6 h, and decreased at 24 h (Figure 1a). However, HSF1 expression was still significantly higher at 24 h.

The same results were confirmed upon in situ hybridization of *HSF1* in zebrafish retina (Figure 1b). A prominent increase in *HSF1* signal was first observed in the ganglion cell layer (GCL) and the inner nuclear layer (INL) at 0.5 h after ONI (Figure 1b). These changes were subsequently enhanced and spread to all of the nuclear layers in the retina, peaking at 6 h after ONI. Similarly, immunohistochemical staining of the HSF1 protein in zebrafish retina detected positive signals in all nuclear layers 1–24 h after ONI (Figure 1c). These increases of HSF1 were accompanied by increases in four heat shock proteins (HSPs), *HSP25, HSP60, HSP70*, and *HSP90*, the target genes of HSF1 examined here by real-time PCR analysis (Appendix A).

### 2.2. Increase in OSK Gene Expression in Zebrafish Retina after ONI

Next, we performed real-time PCR to examine the expression of the three Yamanaka factors—*Oct4, Sox2*, and Klf4 (OSK)—in the injured retina with the optic nerve crushed using gene-specific primers (see Appendix A). Figure 2a showed that the expression of these transcription factors increased significantly and rapidly in the retina within 1 h after ONI. *Klf4* responded most quickly, followed by *Oct4* and then *Sox2*. The localization of OSK was confirmed by in situ hybridization (Figure 2b). In situ hybridization showed the weak expression of Klf4 throughout the retina at 1 h, which then became restricted to the GCL at 3 h after ONI (Figure 2b, upper panel). Meanwhile, *Oct4* showed a prominent signal only in the GCL and INL at 3 h after ONI (Figure 2b, center panel), but this expression then expanded to all nuclear layers at 6 h. The expression of *Sox2* was observed in all nuclear layers including the outer nuclear layer (ONL), INL, and GCL, at 3–6 h after ONI (Figure 2b, lower panel). Immunohistochemical studies of OSK (Appendix A) revealed similar patterns in the real-time PCR and in situ hybridization.

### 2.3. HSF1 Regulates Expression of OSK

We explored the relationship between HSF1 and the three Yamanaka factors, OSK, because the expression of these genes was induced so rapidly in the retina. *HSF1* mRNA expression was increased over 100 times at 6 h after ONI compared with the level in the control (Figure 1a). Therefore, we used morpholino (MO) to suppress *HSF1* expression by the method shown in Figure 3a. Intraocular injection of *HSF1*-specific MO was conducted 20 h before ONI completely suppressed the increase of *HSF1* mRNA in the retina 6 h after ONI (Figure 3b). In addition, treatment with *HSF1* MO completely suppressed the upregulation of *Klf4* mRNA (Figure 3c), *Oct4* mRNA (Figure 3d), and *Sox2* mRNA (Figure 3e). Intraocular injection of standard morpholino (Std. MO) was not effective at suppressing the ONI-induced increase in OSK mRNA (Figure 3b–e).

### 2.4. ChIP Assay of OSK in Response to HSF1

To confirm the correlation between the induction of HSF1 expression after ONI and the subsequent increase in OSK, we performed a ChIP assay by using retinal samples with anti-HSF antibodies. After ONI, DNA samples were extracted from the intact retina (0 h) or injured retina 6 h. These samples were immunoprecipitated with anti-HSF1 antibodies and purified. ChIP-enriched DNA samples were amplified with several primer sets for the encoding of OSK (Appendix A), which have putative HSF1 binding regions (Appendix A). Anti-HSF1 antibodies precipitated approximately 10–20 times more of the specific DNA of each OSK gene than the IgG control did (Figure 4a, 6 h). No amplified products were detected in the IgG control-treated group (Figure 4b, IgG) or the intact group (Figure 4a, 0 h).

## 3. Discussion

### 3.1. Rapid Activation of OSK via HSF1 Signaling in the Zebrafish Retina after ONI

We found that the gene expression of OSK was rapidly induced after ONI. Among these three factors, Klf4 was expressed transiently (1–3 h) and its expression peaked at 1 h. Meanwhile, Oct4 was expressed at 1–6 h and peaked at 3 h. Finally, Sox2 exhibited more long-lasting expression and peaked at 6 h. Regarding the localizations of these molecules, Klf4 was predominantly expressed in the GCL, while Oct4 and Sox2 were first localized in the GCL and INL, and later extended to all nuclear layers including the ONL. Since the gene expression of OSK was so rapidly activated within 1 h after ONI, we tested whether preceding HSF1 directly regulated OSK expression. Injection of the *HSF1*-specific MO into the eye before ONI markedly suppressed OSK gene expression (Figure 3). Furthermore, as the promoter regions of the OSK genes have a consensus sequence that binds to HSF1 (see Appendix A), we performed a ChIP assay. Results of the ChIP analysis with anti-HSF1 antibodies showed that enrichment genomic OSK bind to HSF1. Thus, both the *HSF1* MO treatment assay and the ChIP assay clearly showed that the OSK genes were all regulated by HSF1 (Figure 3 and Figure 4). It is well-known that HSF1 and its target heat shock proteins (HSPs) protect cells under various stresses [22,23,30,31,32,33,34,35,36]. However, a genome-wide study of the biological stress response highlighted novel target genes of HSF1 other than HSPs [30,31]. HSF1, as a master transcription factor, was recently shown to activate many genes related to various cell functions such as development, aging, and carcinogenesis [32,33,34,35]. It is possible that OSK may also be the target of HSF1. Therefore, we concluded that HSF1 directly regulated the gene expression of OSK in the fish retina 1 h after ONI.

### 3.2. Role of Yamanaka Factors in the Injured Retina at the Early Stage of Optic Nerve Regeneration

The serial activations of HSF1-OSK protect cells and maintain their viability after ONI stress. The gene expression of HSF1 started to increase at 0.5 h, peaked at over 100-fold at 6 h, and was still maintained 24 h after ONI. This rapid and widespread retinal expression of HSF1 is essential for cell survival in the acute phase after ONI. When *HSF1*-MO was pre-injected and the optic nerve was injured under conditions of suppressed HSF1 expression, numerous apoptotic cells were observed in all nuclear layers and the retinal layered structure was severely disrupted (Appendix A). The long-lasting HSF1 gene expression must induce the gene expression of cell survival factors such as IGF-1, Bcl-2, and p-Akt 1–5 days after ONI [10,12,14]. Furthermore, the gene expression of OSK was induced rapidly and at the same time after ONI (Figure 2). This rapid activation of Yamanaka factors in the injured retina is also necessary for initializing transformation and maintaining cell survival. Sox2 is a well-established marker of neural stem cells and progenitor cells [37,38,39]. In mammals, Sox2 is highly expressed in the neuroepithelium of the developing central nervous system [39]. In injured zebrafish RGCs, the gene expression of Sox2 could be seen at 1–24 h after ONI (Figure 2 and Appendix A). Retinal neurons have been reported to change their properties in this early stage of optic nerve regeneration. The electrophysiological data reported that spike activities were suddenly lost a few days after ONI [24,40], and then hypertrophic change occurred in fish RGCs [41,42]. These changes indicate that injured RGCs in fish may initiate neural stem cell-like transformation. The other INL and ONL cells also expressed Sox2 strongly, as late as 6 h after ONI (Figure 2 and Appendix A). At present, we think that all retinal neurons transformed into neural stem-like cells under strong HSF1 signals in all nuclear layers at 6 h after ONI. However, future studies are needed to confirm this. Recently, Lu et al. demonstrated that overexpression of the OSK genes in the mouse eye using a viral vector could increase the survival of RGCs, partially regenerate optic axons, and recover vision [43]. This is because the expression of OSK can reset DNA methylation of the gene, allowing the retinal neuron to regain its young state. Interestingly, they also showed that if one of the three OSK factors was missing, the regenerative effect was lost [43]. In our zebrafish retina, the genes encoding these Yamanaka factors were all spontaneously and rapidly activated at the same time of 1 h after ONI, but their expression peaks and durations were slightly different (Figure 2). An in vivo model of OSK activity/expression in the fish retina after ONI would be useful for the next step of addressing the relationship between OSK genes and their target gene expression. The present study clearly showed that the rapid serial activations of HSF1-Yamanaka factors contribute to cell survival and the induction of neuronal stem cells in injured fish retina immediately after ONI.

## 4. Materials and Methods

### 4.1. Animals

Adult zebrafish (Danio rerio; 3–4 cm in length) were used in this study. The zebrafish were anesthetized with 0.02% MS222 (Sigma-Aldrich, St. Louis, MO, USA) in 10 mM phosphate-buffered saline (PBS; pH 7.4). Under anesthesia, the optic nerves on both sides were carefully crushed with forceps 1 mm posterior to the eyeball to create an “injured retina”. Then, the fish were reared in water at 28 °C until the appropriate timepoints. All animal care was performed in accordance with the guidelines for animal experiments of Kanazawa University. Special care was taken to minimize the suffering of the fish.

### 4.2. Tissue Preparation

Retinal samples were prepared for histological analysis at specific timepoints following ONI. Briefly, the eyes were enucleated, bisected, and fixed in 4% paraformaldehyde solution containing 0.1 M phosphate buffer (pH 7.4) and 5% sucrose for 2 h at 4 °C. After infiltration with increasing concentrations of sucrose (5–20%), followed by overnight incubation in 20% sucrose at 4 °C, the tissues were embedded in Optimal Cutting Temperature (OCT) compound (Sakura Fine Technical, Tokyo, Japan) and sectioned at a thickness of 12 µm.

### 4.3. Total RNA Extraction and cDNA Synthesis

Fish were killed by an overdose (0.1%) of MS222 in PBS at appropriate timepoints after ONI. For total RNA extraction, we used Isogen (Nippon Gene, Tokyo, Japan), in accordance with the manufacturer’s instructions. Total RNA samples from each timepoint or treatment were subjected to first-strand cDNA synthesis using a Transcriptor High Fidelity cDNA Synthesis Kit (Roche, Mannheim, Germany).

### 4.4. Quantitative Real-Time PCR

Quantitative real-time PCR was performed with FastStart Essential DNA Probes Master or Green Master Mix (Roche, Mannheim, Germany) using a LightCycler 96 (Roche). On the basis of the zebrafish cDNA sequences (see Appendix A), gene-specific primers were created by Probe Finder using Universal Probe Library (Roche, Mannheim, Germany). The expression levels were analyzed by the ΔΔCt method, using glyceraldehyde 3-phosphate dehydrogenase (GAPDH) as a reference gene. The accession numbers for the genes, DNA sequences of the primer pairs, and lengths of the PCR products used in each experiment are shown in Appendix A.

### 4.5. Immunohistochemistry

Retinal sections from zebrafish were incubated at 121 °C for 10 min in 10 mM citrate buffer. Following washing and blocking, sections were incubated with primary antibodies overnight at 4 °C (HSF1, 1:300; Sox2, 1:500; Oct4, 1:500; Klf4, 1:200). Following incubation with a biotinylated secondary antibody (Vector Laboratories, Burlingame, CA, USA) for 2 h at room temperature, bound antibodies were detected using horseradish peroxidase (HRP)-conjugated streptavidin and 3-amino-9-ethyl carbazole (AEC; Nichirei Biosciences Inc., Tokyo, Japan).

### 4.6. In Situ Hybridization

In situ hybridization was carried out as previously described [15]. Briefly, tissue sections were rehydrated and treated with 5 mg/mL proteinase K (Invitrogen, CA, USA) at room temperature for 5 min. After acetylation and prehybridization, hybridization was performed with cRNA probes labeled with digoxigenin in a hybridization solution overnight at 42 °C. The following day, the sections were washed and treated with 20 mg/mL RNase A at 37 °C for 30 min. To detect the signals, the sections were incubated with an alkaline phosphatase-conjugated anti-digoxigenin antibody (Roche, Rotkreuz, Switzerland) overnight at 4 °C and visualized with tetrazolium-bromo-4-chloro-3-indolylphosphate (Roche) as the substrate.

### 4.7. Chromatin Immunoprecipitation

Chromatin immunoprecipitation (ChIP) was performed using the MAGnifity Chromatin Immunoprecipitation System (Thermo Fisher Scientific, Waltham, MA, USA), in accordance with the manufacturer’s instructions. Briefly, retinal samples were homogenized and linked in 1% formaldehyde for 10 min at room temperature, and 100 mM glycine was added to stop the reaction, followed by washing with cold PBS three times. After centrifugation and ultrasonication using a Bioruptor ultrasonic homogenizer (BM Equipment Co. Ltd., Tokyo, Japan), samples were incubated with magnetic protein A/G beads conjugated with anti-HSF1 (Millipore, CA, USA) or normal IgG, and kept overnight at 4 °C. After immunoprecipitation and washing, the genomic DNA associated with HSF1 was purified and quantified by SYBR Green-based quantitative real-time PCR using the primer sets shown in Appendix A. All primer sets were designed to contain the predicted HSF1 binding region. ChIP dilution buffer was used as a negative control and DNA from the total input was used as an internal positive control.

### 4.8. Intraocular Injection of HSF1 Morpholino to Zebrafish Eye

Vivo-Morpholino (MO) was designed to inhibit the expression of the zebrafish heat shock factor 1 gene via the following sequence: 5′-AGTTTAGTGATGATTTCTGACGGTA-3′. A standard vivo-MO (5′-CCTCTTACCTCAGTTACAATTTATA-3′) was used as a control. All MOs were purchased from GeneTools (Philomath, OR, USA). The MOs were injected into the eye with a Hamilton 33G neuron syringe. Twenty hours after the injection of 0.75 µL of MO solution (0.5 mM) into the eye, the optic nerve was crushed.

### 4.9. Statistical Analysis

To evaluate the mRNA expression of HSF1, Sox2, Oct4, and Klf4, their levels were expressed as the mean ± SEM and the significance of differences was evaluated by one-way ANOVA. Significance was determined at *p* < 0.05 with IBM SPSS Statistic software.

## 5. Conclusions

*HSF1* mRNA was immediately upregulated in the zebrafish retina after ONI. The acute expression of HSF1 directly regulated the expression of Yamanaka factors, which might dedifferentiate retinal neurons at the early stage of optic nerve regeneration after ONI.

## Figures and Tables

**Figure 1 ijms-24-03253-f001:**
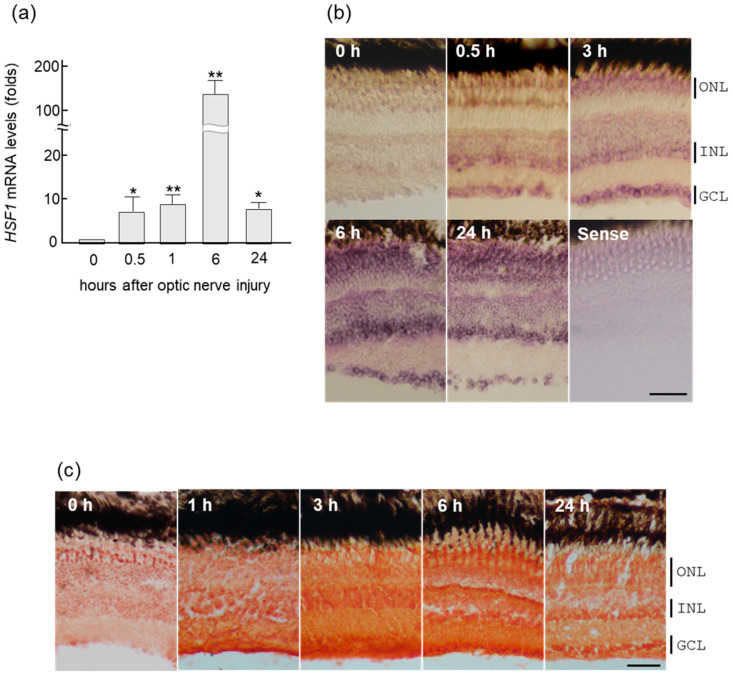
Upregulation of HSF1 (heat shock factor 1) mRNA in zebrafish retina after ONI (optic nerve injury). (**a**) *HSF1* mRNA expression levels after ONI were determined using quantitative real-time PCR. (**b**) In situ hybridization of *HSF1* in the zebrafish retina after nerve injury. *HSF1* mRNA started to increase in the retina for 0.5 h and peaked at 6 h after ONI. Its localization was first seen in the GCLs (ganglion cell layers) and after the INLs (inner nuclear layers). Then, these signals spread to all nuclear layers including the ONLs (outer nuclear layers) at 6 h and slightly decreased at 24 h after ONL. (**c**) Immunohistochemical staining of HSF1 in the zebrafish retina after ONI. Significant immunostaining peaked at 3 to 6 h in all nuclear layers after ONI. Data are expressed as the mean ± SEM of five independent experiments and analyzed by one-way ANOVA, followed by Scheffe’s multiple comparisons. Statistical significance was set at * *p <* 0.05 or ** *p <* 0.01. Scale bar = 50 μm.

**Figure 2 ijms-24-03253-f002:**
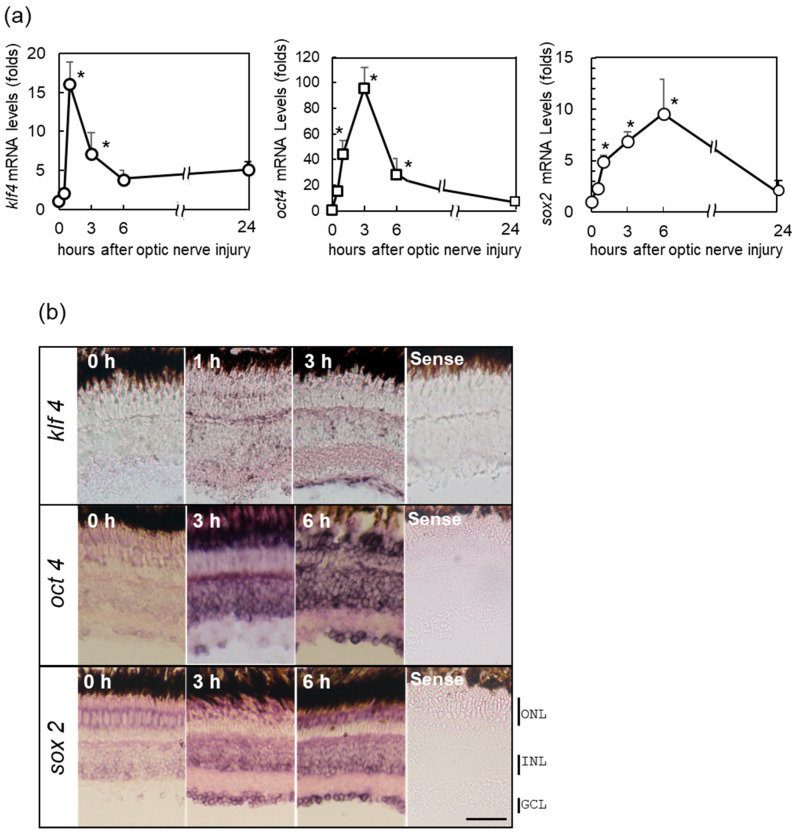
Upregulation of the Yamanaka factors (OSK) in zebrafish retina after ONI. (**a**) mRNA expression levels of OSK after ONI were determined by quantitative real-time PCR (left, klf4; center, oct4; right, sox2). (**b**) In situ hybridization of OSK in zebrafish retina after ONI. *Klf4* mRNA expression started to increase at 1 h and localized to the GCLs at 3 h after ONI. *Oct4* mRNA signal was observed in the GCL and strongly in the INL and ONL at 3 h, but this strong signal was seen in all nuclear layers at 6 h after ONI. *Sox2* mRNA expression was observed in all nuclear layers 3 h after ONI and more prominent at 6 h. No positive signals could be seen with the sense probe (Sense). Five to six experiments were repeated with different retinas under each experimental condition and produced the same results. Data are expressed as the mean ± SEM and analyzed by one-way ANOVA, followed by Scheffe’s multiple comparisons. Statistical significance was set at * *p* < 0.05. Scale bar = 50 μm.

**Figure 3 ijms-24-03253-f003:**
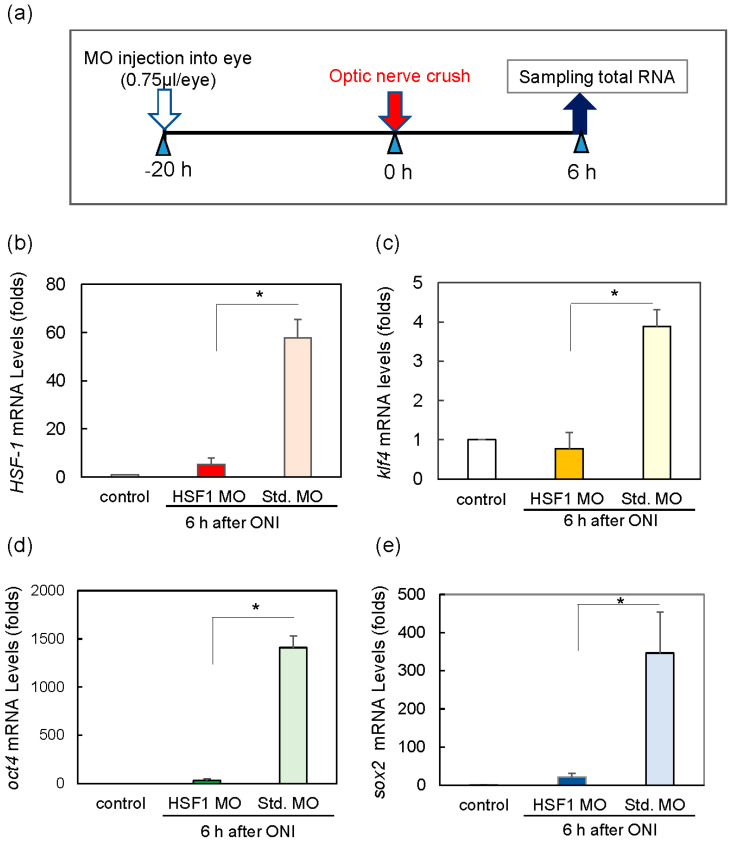
Treatment of HSF1 MO (morpholino) significantly reduced the mRNA expression of Klf4, Oct4, and Sox2 6 h after ONI. (**a**) HSF1 MO or standard MO (Std. MO) was injected intraocularly 20 h before ONI. (**b**) HSF1 MO-treated group suppressed HSF1 mRNA expression compared to the Std. MO-treated group. Under these conditions, the mRNA expression of Klf4 (**c**), Oct4 (**d**), and Sox2 (**e**) was inhibited compared to the control (Std. MO) groups. Five experiments were repeated under each experimental condition. Data are expressed as the mean ± SEM of independent experiments and analyzed by one-way ANOVA, followed by Scheffe’s multiple comparisons. Statistical significance was set at * *p* < 0.05.

**Figure 4 ijms-24-03253-f004:**
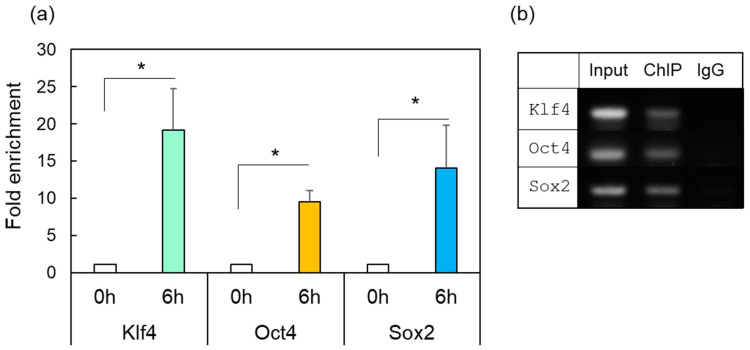
ChIP-enriched DNA was prepared using preimmune serum (IgG) or anti-HSF1 antibody from the control (0 h) or damaged zebrafish retina after ONI (6 h). (**a**) The immunoprecipitated DNA of *Klf4*, *Oct4*, and *Sox2* were analyzed by real-time PCR. Each ChIP signal was divided by the no-antibody signals (IgG), representing the ChIP signals as the fold increase in signals relative to the background signals. (**b**) Gel electrophoresis image using the ChIP samples. The input was used as an internal positive control for the ChIP assay. Five to six experiments were repeated with different retinas under each experimental condition. Data are expressed as the mean ± SEM of independent experiments and analyzed by one-way ANOVA, followed by Scheffe’s multiple comparisons. Statistical significance was set at * *p* < 0.05.

## Data Availability

Not applicable.

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
