# Peer review of "Specific Activation of Yamanaka Factors via HSF1 Signaling in the Early Stage of Zebrafish Optic Nerve Regeneration"

_ijms, 2023, doi:10.3390/ijms24043253_

Round 1
Reviewer 1 Report
The manuscript “Specific activation of Yamanaka factors via HSF1 signaling in the early stage of zebrafish optic nerve regeneration” by Sugitani et al. is devoted to establishing factors involved in optic nerve regeneration. In particular, they investigated the expression of Yamanaka factor genes after the optic nerve injury in zebrafish and their relationship with HSF1 expression.
The work is well-written and can be of interest to a broad scientific audience. It can be published if deficiencies are addressed.
Deficiencies:
1. Most of the references are pretty outdated. Therefore, I would recommend updating references.
Minor deficiencies:
2. How many fish were used in the experiments to generate each data point?
3. Section 4.9: Which tool was used for statistical analysis?
4. Line 52-54: “In consideration of the similarity between the regenerative process of the fish optic nerve and the normal development of the visual system in embryogenesis as mentioned above,…” The authors mentioned embryogenesis, as discussed already. However, it is not the case. Please add a discussion or rephrase this sentence.
5. Line 63: There are certain terminology irregularities. In lines 59-60, the authors refer to Yamanaka factors as a 4-factor model: “The term “Yamanaka factors” originally referred to four transcription factors, Oct4, Sox2, Klf4, and c-Myc”. Then, they refer to 3 factors (Oct4, Sox2, Klf4) as OSK. Firstly, the abbreviation of OSK should be explicitly introduced. It can be implied from the text; however, it is unclear. Secondly, in the abstract (lines 17-18), the authors refer to the OSK subset of Yamanaka factors as all Yamanaka factors “mRNA expression of all Yamanaka factors (Oct4, Sox2, and Klf4: OSK)”, which is confusing.
6. Line 158: Captions to Fig 3 should follow Fig 3. There is a text (lines 153-157) between them.
7. Line 177: Captions to Fig 4 should follow Fig 4. There is a text (lines 171-174) between them.
8. Line 167: In Methods, the authors describe the procedure as crashing the optic nerve: “the optic nerve was carefully crushed with forceps 1 mm posterior to the eyeball” (line 243). However, in the body of the article (line 167), they refer to the intact and injured retina.
9. Line 285: Incorrect reference type: “(Sugitani et al., 2012)”
Author Response
Dear Reviewer 1,
Thank you very much for your editorial work on our manuscript entitled “Specific activation of Yamanaka factors via HSF1 signaling in the early stage of zebrafish optic nerve regeneration” (Manuscript ID: ijms-2163107).
We have revised it according to your comments and suggestions. The followings are the specific suggestions and our replies.
- Most of the references are pretty outdated. Therefore, I would recommend updating references.
-> References were checked again and new lists were added.
Minor deficiencies:
- How many fish were used in the experiments to generate each data point?
->The number of zebrafish was added to each figure legend.
3. Section 4.9: Which tool was used for statistical analysis?
->Significance was determined at p < 0.05 with IBM SPSS Statistic software.
This sentence was added on line 301.
- Line 52-54: “In consideration of the similarity between the regenerative process of the fish optic nerve and the normal development of the visual system in embryogenesis as mentioned above,…” The authors mentioned embryogenesis, as discussed already. However, it is not the case. Please add a discussion or rephrase this sentence.
->We have rewritten it as follows
“In consideration of the similarity between the regenerative process of the fish optic nerve and the normal development of the visual system in embryogenesis”
- Line 63: There are certain terminology irregularities. In lines 59-60, the authors refer to Yamanaka factors as a 4-factor model: “The term “Yamanaka factors” originally referred to four transcription factors, Oct4, Sox2, Klf4, and c-Myc”. Then, they refer to 3 factors (Oct4, Sox2, Klf4) as OSK. Firstly, the abbreviation of OSK should be explicitly introduced. It can be implied from the text; however, it is unclear. Secondly, in the abstract (lines 17-18), the authors refer to the OSK subset of Yamanaka factors as all Yamanaka factors “mRNA expression of all Yamanaka factors (Oct4, Sox2, and Klf4: OSK)”, which is confusing.
->In response to your suggestion, we have modified the text as follows
Abstracts Line15:
Here, we focused on the expression of Yamanaka factors, which are well-known…
-> Here, we focused on the expression of three Yamanaka factors (Oct4, Sox2, and Klf4: OSK), which are well-known…
Line17-18:
mRNA expression of all Yamanaka factors was …
-> mRNA expression of OSK was…
- Line 158: Captions to Fig 3 should follow Fig 3. There is a text (lines 153-157) between them.
-> Arrangement of figures and figure descriptions have been corrected.
- Line 177: Captions to Fig 4 should follow Fig 4. There is a text (lines 171-174) between them.
->Arrangement of figures and figure descriptions have been corrected.
- Line 167: In Methods, the authors describe the procedure as crashing the optic nerve: “the optic nerve was carefully crushed with forceps 1 mm posterior to the eyeball” (line 243). However, in the body of the article (line 167), they refer to the intact and injured retina.
->The optic nerve is a bundle of axons of retinal ganglion cells (RGC) in the retina. Thus, an optic nerve-damaged retina is often described as an injured retina. To avoid misinterpretation, Methods 4.1 line 224 was changed as follows.
the optic nerve was carefully crushed with forceps 1 mm posterior to the eyeball
→the optic nerves on both sides were carefully crushed with forceps 1 mm posterior to the eyeball to create “injured retina”
- Line 285: Incorrect reference type: “(Sugitani et al., 2012)”
I have rewritten it as follows
-> In situ hybridization was carried out as previously described [15].

Reviewer 2 Report
The manuscript by Sugitani et al shows that HSF1 is required for OSK expression after optic nerve injury. The topic although not very novel, similar ideas have been explored (e.g. PMID: 30884523), it is of broad interest to the ophthalmology field. However, the manuscript show limited data and lacks several control experiments, being required further data to substantiate this work.
Major:
1. The lack of controls is dramatic. Authors should perform the crush in one eye and use the other as co-lateral control. It is not clear if, in the 0H timepoint, the crush of the optic nerve is performed or not. If so the author should have another group of control (sham) eyes where the optic nerve is not crushed.
2. In the in situ experiments are missing a control with the sense probe to assess the possible unspecific signal.
3. OCT4 qPCR and ISH do not match. Please elaborate on that.
4. Authors should demonstrate that HSF1 protein is ablated 20H after morpholine injection.
5. What are the effects of HSF1 knockdown in the retina? Authors should explore the possible consequences of HSF1 knockdown on the retina.
6. Are client HSP affected by the reduction of HSF1 and can these influence the recruitment of OSK?
7. Although authors focused only on early timepoints it would be interesting to see the long-term effect of HSF1 deletion on Ganglion cell regeneration.
8. How nerve optic nerve injury induces full retinal expression of HSF1? Authors should speculate on that.
Minor:
- Please make clear that the fold differences are relative to 0H timepoint.
- Fig1: Please add the IHC pictures to figure 1. Please enlarge the pictures on B.
- Authors should discuss HSF1 expression by retinal cells. Is it expressed in all the retinal cells?
- line 190, the statement about klf4 is not supported by the data.
- Line 225-227: the sentence is confusing, it is not clear the message that the authors want to transmit.
- Line 244: the procedure is done in both eyes?
Author Response
Dear Reviewer 2,
Thank you very much for your editorial work on our manuscript entitled “Specific activation of Yamanaka factors via HSF1 signaling in the early stage of zebrafish optic nerve regeneration” (Manuscript ID: ijms-2163107).
We have revised it according to your comments and suggestions. The followings are the specific suggestions and our replies.
Major:
- The lack of controls is dramatic. Authors should perform the crush in one eye and use the other as co-lateral control. It is not clear if, in the 0H timepoint, the crush of the optic nerve is performed or not. If so the author should have another group of control (sham) eyes where the optic nerve is not crushed.
->Both eyes were operated on to eliminate the possibility of postoperative "sympathetic ophthalmitis" or "sympathetic ophthalmia" (PMID: 28414613, PMID: 5119708 etc.). Therefore, both optic nerves were crushed similarly. “0h” is intact control in which the optic nerve is not crushed.
- In situ experiments, a control with the sense probe is missing to assess the possible unspecific signal.
->We added sense probe data in figure1b and figure 2b.
No positive signals could be seen with the sense probe.
- OCT4 qPCR and ISH do not match. Please elaborate on that.
You are correct in pointing this out. We do not know the cause.
However, the immunostaining results (Figure S2) are very well linked and support the real-time PCR results (Figure 2A). Therefore, we re-selected another ISH image that matches well with these results (Figure 2B, Oct4 3h).
- Authors should demonstrate that HSF1 protein is ablated 20H after morpholine injection.
Please see the figure below. Immunohistochemical staining of HSF1 protein in the zebrafish retina after morpholino injection. If necessary, attach it as a supplemental figure.
- MO injection (Left panel): 20 hrs after HSF1-MO injection. → Same as IHC results for intact retina “0h” (see New Figure 1c)
- Optic nerve crush + Std-MO injection 20 hrs before (central)→Positive staining of HSF1 detected all nuclear layers compared to the MO injection group.
- Optic nerve crush + HSF1-MO injection 20 hrs before (Right)→The expression of HSF1 protein was lost, and the retinal layers underwent apoptosis and did not retain normal structure.
- What are the effects of HSF1 knockdown in the retina? Authors should explore the possible consequences of HSF1 knockdown on the retina.
→When HSF1 was knockdown by HSF1-MO injection, no tissue degeneration was observed in the retina (upper Figure: MO injection). However, when the retina was damaged by optic nerve crush, apoptosis was induced immediately and the retina was unable to maintain its layered structure within 24 hours (upper Figure: HSF1-MO injection+ONI).
Thus, HSF1 is supported to play an important role in protecting retinal cells after optic nerve injury and ensuring their continued survival.
- Are client HSP affected by the reduction of HSF1 and can these influence the recruitment of OSK?
→Optic nerve injury induced not only HSF1 gene expression but also HSPs (HSP25, HSP60, HSP70, HSP90) upregulation (Figure S1, Results line82-84). Gene expression of OSK must be preceded by HSF1 expression as well (Figure 3). However, the present study does not confirm a direct link between HSPs and OSK. We believe it is via HSF1 signaling.
- Although authors focused only on early time points it would be interesting to see the long-term effect of HSF1 deletion on Ganglion cell regeneration.
→ HSF1 changes were very rapid, peaking at 6 hours after optic nerve injury. Therefore, we focused on changes limited to the first 24 hours after optic nerve injury. In the future, we would like to consider perspectives on the long-term effects of HSF1 deletion on optic nerve regeneration.
- How nerve optic nerve injury induces full retinal expression of HSF1? Authors should speculate on that.
→I'm not sure at this point how that is. Previous studies have shown that optic nerve injury does not alter RGCs, but induces molecules expressed only in photoreceptors (PMID: 15385617). Therefore, we think that signals from optic nerve damage are immediately transmitted throughout the retina. What induces HSF1 expression is unknown, it’s the next question. However, after optic nerve injury, HSF1 expression is essential for neuronal survival in the retina.
The figure below shows the detection of FITC-labeled apoptotic cells in the retina after ONI in which HSF1 knockdown by MO injection.
In the retina injected with HSF1-MO, numerous apoptotic signals can be observed in all nuclear layers. On the other hand, in retinas injected with control MO (Std. MO), no apoptotic cells can be seen. Thus, it is presumed that HSF1 expression is vital for cell survival in the retina after ONI.
Minor:
- Please make clear that the fold differences are relative to 0H timepoint.
→“0 h” is intact control in which the optic nerve is not crushed. Each data was compared to this control (0h).
- Fig1: Please add the IHC pictures to figure 1. Please enlarge the pictures on B.
→HSF1 IHC staining results were added in Figure 1c.
Figure 1b enlarged in size.
- Authors should discuss HSF1 expression by retinal cells. Is it expressed in all the retinal cells?
→From the results of histological experiments, including immunohistochemistry and in situ hybridization, it was concluded that HSF1 is expressed in all nuclear layers in the retina after optic nerve injury. The following sentences were added to “Discussion” Line 209-.
“This rapid and widespread retinal expression of HSF1 is essential for cell survival in the acute phase after optic nerve injury. When HSF1-MO was pre-injected and the optic nerve was injured under conditions of suppressed HSF1 expression, numerous apoptotic cells were observed in all nuclear layers and the layered structure of the retina was severely disrupted (data not shown).”
- line 190, the statement about klf4 is not supported by the data.
→Figure 2a, the results of real-time PCR showed that gene expression of klf4 was so rapidly activated and peaked at 1 h after ONI.
- Line 225-227: the sentence is confusing, it is not clear the message that the authors want to transmit.
→We added the text as follows in line 218,
This is because the expression of OSK can reset DNA methylation of the gene, allowing the retinal neuron to regain its young state.
- Line 244: the procedure is done in both eyes?
→We have rewritten the text as follows,
Line 236
Under anesthesia, the optic nerves on both sides were carefully crushed with forceps 1 mm posterior to the eyeball.

Round 2
Reviewer 2 Report
The authors replied adequately to most of my previous comments. However, they decided not to include data related to later time points in the manuscript.
Minor:
Data regarding the statement below, the authors should be quantified and added to the manuscript
"This rapid and widespread retinal 196 expression of HSF1 is essential for cell survival in the acute phase after ONI. When 197 HSF1-MO was pre-injected and the optic nerve was injured under conditions of sup-198 pressed HSF1 expression, numerous apoptotic cells were observed in all nuclear layers 199 and the retinal layered structure was severely disrupted (data not shown). "
Author Response
Dear Reviewer 2,
Thank you very much for your editorial work on our manuscript entitled “Specific activation of Yamanaka factors via HSF1 signaling in the early stage of zebrafish optic nerve regeneration” (Manuscript ID: ijms-2163107).
We have revised it according to your comments and suggestions.
The followings are the specific suggestions and our replies.
Data regarding the statement below, the authors should be quantified and added to the manuscript
"This rapid and widespread retinal 196 expression of HSF1 is essential for cell survival in the acute phase after ONI. When HSF1-MO was pre-injected and the optic nerve was injured under conditions of suppressed HSF1 expression, numerous apoptotic cells were observed in all nuclear layers and the retinal layered structure was severely disrupted (data not shown). "
→We have added a new Figure S4 with your suggestion.
Line 201 data not shown→Figure S4

Round 3
Reviewer 2 Report
authors replied to my last comments.